# *FLT3*-ITD Expression as a Potential Biomarker for the Assessment of Treatment Response in Patients with Acute Myeloid Leukemia

**DOI:** 10.3390/cancers14164006

**Published:** 2022-08-19

**Authors:** Diego Carbonell, María Chicano, Alfonso J. Cardero, Ignacio Gómez-Centurión, Rebeca Bailén, Gillen Oarbeascoa, Diana Martínez-Señarís, Carolina Franco, Paula Muñiz, Javier Anguita, Mi Kwon, José Luis Díez-Martín, Ismael Buño, Carolina Martínez-Laperche

**Affiliations:** 1Department of Hematology, Gregorio Marañón General University Hospital, 28007 Madrid, Spain; 2Hematological Genetics Lab, Gregorio Marañón Health Research Institute (IiSGM), 28007 Madrid, Spain; 3Department of Hematology, A Coruña University Hospital Complex, 15006 A Coruña, Spain; 4Department of Medicine, School of Medicine, Complutense University of Madrid, 28040 Madrid, Spain; 5Genomics Unit, Gregorio Marañón General University Hospital, IiSGM, 28007 Madrid, Spain; 6Department of Cell Biology, School of Medicine, Complutense University of Madrid, 28040 Madrid, Spain

**Keywords:** *FLT3*-ITD gene expression, acute myeloid leukemia, allogeneic stem cell transplantation, tyrosine kinase inhibitor

## Abstract

**Simple Summary:**

*FLT3*-internal tandem duplication (ITD) mutation analysis in DNA samples is essential for optimal clinical management in patients with acute myeloid leukemia (AML). However, the utility of *FLT3*-ITD mutation analysis in cDNA samples and its use as a follow-up biomarker are controversial. In this context, we compared mutation analyses between DNA and cDNA samples and evaluated the use of cDNA analysis for AML monitoring. *FLT3*-ITD mutation analysis in cDNA samples demonstrated a higher sensitivity than those in DNA samples. In particular, in patients undergoing allogeneic hematopoietic stem cell transplantation, the disease was detected long before they relapsed. In addition, cDNA analysis evaluated the patients’ response to FLT3 inhibitors. Therefore, *FLT3*-ITD cDNA could be a useful additional biomarker in patients with AML, for both the diagnosis and for assessing the treatment response.

**Abstract:**

*FLT3*-internal tandem duplication (ITD) analysis is not typically performed in cDNA samples and is not considered an appropriate marker for monitoring measurable residual disease (MRD). The aims of this study were to compare *FLT3*-ITD mutation analysis in DNA and cDNA samples at diagnosis and to demonstrate the usefulness of its expression measurement as an MRD marker after allogeneic stem cell transplantation (allo-HSCT) or FLT3 inhibitor (FLT3i) administration. A total of 46 DNA and cDNA diagnosis samples, 102 DNA and cDNA post-allo-HSCT samples from 34 patients and 37 cDNA samples from 7 patients with refractory/relapse AML treated with FLT3i were assessed for the *FLT3*-ITD mutation through fragment analysis. In terms of sensitivity, the analysis of cDNA was superior to that of DNA, quantifying higher allelic ratio values in most cases at diagnosis, and thus optimizing the detection of minor clones and prognostic classification. Regarding the last sample before post-HSCT relapse, cDNA analysis anticipated relapse in most cases, unlike DNA analyses. With regard to the post-FLT3i follow-up, *FLT3*-ITD expression was reduced after the first FLT3i cycle when the treatment was effective, whereas it was not reduced in refractory patients. *FLT3*-ITD expression could be a useful additional biomarker at diagnosis and for the assessment of MRD after allo-HSCT and FLT3i in AML.

## 1. Introduction

The *FLT3* gene encodes a receptor tyrosine kinase involved in important homeostatic processes, such as the survival, differentiation and proliferation of hematopoietic stem cells [1]. *FLT3*-internal tandem duplication (ITD), which activates its protein function constitutively, is one of the most common gene mutations found in patients with acute myeloid leukemia (AML) [2]. This mutation plays a decisive role in AML prognosis based on European Leukemia Net (ELN) guidelines, and leads to poor outcomes. For this reason, *FLT3*-ITD mutation screening is essential for the optimal clinical management of patients [3,4]. Allogeneic hematopoietic stem cell transplantation (allo-HSCT) is one of the treatments with a high cure rate in high-risk cases of AML; however, relapse after allo-HSCT remains a problem in these patients [5]. Hence, the importance of the FLT3 protein as a target for achieving and maintaining complete remission has increased in recent years [6,7]. To this end, several *FLT3* inhibitors (FLT3i) have been developed. FLT3i are divided into two groups: type I FLT3i, such as midostaurin [8] or gilteritinib [9], and type II, including sorafenib [10] or quizartinib [11]. While type I inhibitors can interact with either active or inactive *FLT3* conformations, type II inhibitors only bind to the inactive form [12]. Regarding *FLT3*-ITD mutation detection, the standard screening methodology is PCR followed by capillary electrophoresis for fragment length analysis. Its quantification is calculated by dividing the area under the curve (AUC) of the mutant allele by the AUC of the wild-type allele, expressed as the allelic ratio (AR) value [3]. Regarding the type of sample, DNA is typically preferred to cDNA for detecting and quantifying the AR of *FLT3*-ITD [3]. Although ITD mutation screening in *FLT3* is mandatory at diagnosis, its use as a genetic marker for measurable residual disease (MRD) remains controversial due to the poor sensitivity of its detection method and its apparent instability throughout the course of the disease [13,14]. In this context, the first objective of this study was to assess the main differences in the quantification of the *FLT3-ITD* mutation between DNA and cDNA samples and to detect whether those differences have an impact on the diagnosis of the disease. The second aim was to assess the usefulness of measuring its expression as an MRD marker after treatment with FLT3i and post-allo-HSCT.

## 2. Materials and Methods

### 2.1. Patient Samples

Forty-six patients with newly diagnosed *FLT3*-mutated AML were consecutively included and studied at diagnosis. Thirty-four patients with *FLT3*-mutated AML in their first complete remission treated with allo-HSCT in our center (diagnosed in our center or derived from another hospital) were also consecutively included and studied at various time points. Twelve of these patients subsequently relapsed and one lost the *FLT3*-ITD mutation at relapse. Lastly, seven patients with refractory/relapsed (R/R) *FLT3*-mutated AML treated with FLT3i were included and studied at various time points. The ethics committee of the Gregorio Marañón General University Hospital approved the study (No. 03/201503, 3 March 2015) and all patients signed the informed consent document.

In order to evaluate the usefulness of *FLT3*-ITD expression measurement by capillary electrophoresis at distinct time points, 59 patients were divided into three cohorts: diagnosis, post-allo-HSCT and post-FLT3i (with 46, 34 and 7 patients, respectively; Figure 1, Table 1). The diagnosis cohort was analyzed to compare the *FLT3*-ITD mutation ratio between DNA and cDNA (Appendix A). Follow-up cohorts were analyzed to evaluate the usefulness of DNA and cDNA *FLT3*-ITD mutation ratio measurement as a biomarker for MRD. 

### 2.2. Sample Processing

A total of 185 samples were collected in EDTA tubes from 59 patients, 160 from bone marrow (BM) and 25 from peripheral blood (PB). Forty-six diagnosis samples were obtained from 46 patients. A total of 102 samples from 34 patients who underwent allo-HSCT were obtained: 34 samples pre-allo-HSCT, 34 samples at day 30 after infusion and, from those patients who relapsed, 22 samples before relapse and 12 relapse samples. Lastly, 37 samples from 7 patients treated with FLT3i were obtained before the first cycle and after every cycle. DNA was purified from samples using a Maxwell RSC Blood DNA Kit (Promega, Madison, WI, USA). RNA was isolated using TRIzolTM reagent (Invitrogen, Waltham, MA, USA) and cDNA synthesis through reverse-transcription was performed using the First Strand cDNA Synthesis kit (Roche, Basel, Switzerland). 

### 2.3. FLT3-ITD Mutation Analysis and Quantification

*FLT3*-ITD mutations were analyzed in DNA and cDNA samples by fragment analysis through PCR with subsequent capillary electrophoresis. PCR was performed for both DNA and cDNA using specific primers (forward primer-FW-5′-GGTGTCGAGCAGTACTCTAAACATGAGTG-3′ and reverse primer-RV-5′-6FAM-GATCCTAGTACCTTCCCAAACTC-3′ for DNA PCR; and FW 5′-AGCAATTTAGGTATGAAAGCCAGCTA-3′ and RV 5′-6FAM-CTTTCAGCATTTTGACGGCAACC-3′ for cDNA PCR) under the following conditions: 95 °C for 9 min, 35 cycles at 98 °C for 30 s, 56 °C for 1 min, 72 °C for 2 min and 72 °C for 7 min using a Veriti™ 96-Well Fast Thermal Cycler (Thermo Fisher, Waltham, MA, USA). Capillary electrophoresis was performed in an ABI3130xl DNA sequencer (Applied Biosystems, Waltham, MA, USA). Fragment analysis was performed with Peak Scanner™ Software 2.0 (Thermo Fisher, USA). The AR was quantified by dividing the AUC of the mutant allele by the AUC of the wild-type allele. AMLs with allelic ratios ≥0.5 were classified as high-risk AML according to the ELN 2017 risk stratification algorithm. Cut-off values for *FLT3*-ITD AR by capillary electrophoresis were 0.03 for diagnosis samples and 0.01 for MRD samples.

### 2.4. Measurable Residual Disease and Chimerism Analysis

MRD was followed up with *NPM1* or *WT1* RT-qPCR; *NPM1* expression was analyzed after allo-HSCT by RT-qPCR in those patients with *NPM1*-mutated AML. In the patients without *NPM1* mutations, *WT1* expression was measured through RT-qPCR after allo-HSCT. All RT-qPCR tests were performed in a LightCycler 1.5 (Roche, Switzerland) using specific probes and primers, as previously described [15,16]. MRD was also analyzed by multiparametric flow cytometry (MFC) in every BM sample during follow-up using monoclonal antibodies corresponding to the immunophenotypic profile identified in the diagnosis of each patient, using a DxFLEX cytometer (Beckman Coulter, Brea, CA, USA).

Chimerism analysis by STR-PCR was performed using the AmpFlSTR™ SGM Plus™ PCR Amplification Kit (Applied Biosystems) and the Mentype^®^Chimera^®^ kit (Biotype, Dresden, Germany), which included polymorphic, autosomal, non-coding STR loci (10 for SGM Plus™ and 12 for Chimera^®^) and Amelogenin as a sex-specific marker. PCR was carried out using specific primers fluorescence-labeled with 6-FAM^TM^, BTG or BTY in a GeneAmp^®^ PCR System 9700 Thermal Cycler (Applied Biosystems, USA), followed by capillary electrophoresis using an ABI3130xl DNA sequencer (Applied Biosystems). Electropherograms were analyzed through GeneMapper™ Software 5 (Thermo Fisher, USA).

### 2.5. Statistical Analysis

Categorical variables were expressed as frequencies and percentages and quantitative variables were expressed as medians and range. In terms of comparison between DNA and cDNA *FLT3*-ITD mutation AR, the Wilcoxon test was performed for associations and the Spearman test for correlations. For the comparison of diagnostic samples in those patients carrying more than one *FLT3*-ITD mutation, every mutation was analyzed as an event, comparing its DNA ratio with its cDNA ratio. Statistical significance was set at *p* < 0.05 and all statistical tests were two-sided. Analyses were performed with R software (3.3.2 version) and graphs were carried out using GraphPad Prism 7.

## 3. Results

### 3.1. FLT3-ITD Mutation in DNA and cDNA in Diagnostic Samples (Allelic Ratio)

The *FLT3*-ITD mutation analysis was performed in both DNA and cDNA samples at diagnosis from 46 patients carrying the mutation. Thirty-one patients (67%) presented one mutation (single clone), fourteen patients (31%) carried two clones and one patient (2%) had three clones. In total, 62 mutations were detected. One secondary clone was not detected in the DNA sample after having been identified in cDNA. Four mutations from two patients were excluded from the analysis due to the lack of a wild-type *FLT3* allele in the cDNA sample, which limited the AR calculation. With regard to the AR comparison between the sample types, Spearman’s test revealed a correlation coefficient of 0.85 (0.78–0.92). The median burden of the 58 analyzed *FLT3*-ITD mutations was 0.54 (range 0–9.47) in the DNA samples and 0.63 (range 0.01–13) in the cDNA samples, a statistically significant difference (Wilcoxon test, *p* < 0.001, Figure 2). In 40 patients (87%), the prognosis based on the ELN 2017 risk stratification algorithm did not change due to AR, whereas, in 6 patients (13%), the *FLT3*-ITD mutation burden was <0.5 in DNA and ≥0.5 in cDNA, which changed their risk stratification. One patient was not a candidate for chemotherapy and, of the remaining five, four were refractory or relapsed after intensive treatment.

### 3.2. FLT3-ITD Mutations in DNA and cDNA for Post-Allo-HSCT Monitoring

A comparison of *FLT3*-ITD mutation detection between DNA and cDNA samples was also performed during the follow-up of 34 patients who underwent allo-HSCT. Regarding the pre-allo-HSCT sample, seven patients (21%) were positive for *FLT3*-ITD mutation in cDNA and two (6%) in DNA samples; only one of the cases was coincident (Appendix A). Among the seven patients who were positive in cDNA, four relapsed after allo-HSCT, three of whom presented AR > 0.1 pre-allo-HSCT. The three patients who did not relapse had a *FLT3*-ITD allelic burden between 0.03 and 0.05. Regarding the two patients who were positive in DNA samples, one relapsed after allo-HSCT and their mutation was also detected in cDNA with a high allele ratio. The remaining patient, in whom the mutation was only detected in the DNA sample, did not relapse. Eleven and eight patients who relapsed were negative for *FLT3*-ITD mutations in DNA and cDNA before allo-HSCT, respectively. None of the patients were positive on day 30 after allo-HSCT.

With respect to the last sample before relapse (obtained a median of 22 days before relapse, range 7–85), five out the twelve cases (42%) were positive for *FLT3*-ITD mutations in DNA samples and nine (75%) were positive in cDNA samples, with significant differences observed between the AR of both types of samples (Wilcoxon test, *p* < 0.001, median [range] 0 [0–0.04] vs. 0.15 [0–0.57], Figure 3). Regarding the three cases that were negative in cDNA, one patient lost the *FLT3*-ITD mutation at relapse and the other two samples were also negative for *NPM1* or *WT1* RT-qPCR and were in complete donor chimerism analysis at that time. When considering the entire follow-up monitoring analysis, of the 16 samples that were positive in cDNA (with a median of 37 days before relapse, range 14–97), 13 were negative in DNA and 12 were in complete donor chimerism. With respect to the RT-qPCR analysis, in all cases but one, *NPM1* or *WT1* RT-qPCR was positive when *FLT3*-ITD mutation was detected in cDNA (Figure 4). In that case (patient 4), a *FLT3*-ITD mutation was detected in cDNA with an AR of 0.02, and the *WT1* RT-qPCR was negative (Figure 4).

### 3.3. FLT3-ITD Expression for FLT3i Treatment Monitoring

Given that the measurement of *FLT3*-ITD mutations seemed to have a higher sensitivity when analyzing cDNA samples, we decided to evaluate whether it could be used as an MRD marker for monitoring patients with R/R *FLT3*-mutated AML treated with FLT3i. The median *FLT3*-ITD mutation ratio was 1.09 (range 0–5.3). In all but one case (patient 6), *FLT3*-ITD mutation levels increased or were maintained during follow-up, indicating refractoriness to inhibitors. Patient 6, who was treated with gilteritinib, achieved complete remission and an absence of *FLT3*-ITD expression until after nine cycles of FLT3i (Figure 5).

## 4. Discussion

The *FLT3*-ITD mutation status is one of the most important variables to consider when defining the prognosis and treatment of patients with AML [17]. Although the current standard technique recommended for genetic analysis in patients with AML is next-generation sequencing (NGS), it is not globally available, and capillary electrophoresis is widely used for both detecting *FLT3*-ITD mutations and determining the AR. Moreover, its methodological simplicity and its prompt results facilitate the early management of AML. Therefore, it is recommended to determine the *FLT3* mutational status promptly after diagnosis to optimize clinical management [3,17]. However, this analysis is typically performed on DNA samples, and the potential usefulness of *FLT3*-ITD expression as a biomarker has not yet been fully studied. In this regard, the present study focused on elucidating its utility both at diagnosis and for the monitoring of patients after allo-HSCT or FLT3i treatment. The analysis of cDNA samples by capillary electrophoresis to detect *FLT3*-ITD mutations at diagnosis has been described in a few studies, mainly in pediatric patients [18,19] and, to the best of our knowledge, in only one study in adult patients [20]. There is controversy regarding whether to use DNA or cDNA for this analysis. While AML guidelines recommend that *FLT3*-ITD mutation analyses be performed on DNA samples [3,4], some authors argue that its expression assessment is suitable for genetic analysis. These studies assert that, in addition to being more sensitive, cDNA analysis is almost equivalent to DNA in terms of AR measurement. Furthermore, they report not only risk stratification changes when the mutation was analyzed in cDNA in several patients but also a more accurate classification of their outcome [19]. In the present study, significant differences between DNA and cDNA AR quantification were observed, even to the point of changing the prognosis of some patients who had mostly unfavorable outcomes. cDNA was more sensitive than DNA, finding higher AR values in most cases. No wild-type allele was detected in the cDNA samples of two patients. Although the wild-type alleles were detected in DNA, there appeared to be no normal allele expression in these two cases, which would have a similar effect to a loss of heterozygosity. A loss of the wild-type allele has been associated with poorer overall survival and progression-free disease [21]. In the case of these two patients, both had an unfavorable clinical course, confirming this prognosis. In addition, in one patient who ultimately relapsed with *FLT3*-ITD, the mutation was only detected at diagnosis in cDNA, emphasizing the usefulness of the cDNA approach. Our results suggest that the analysis of cDNA samples at diagnosis could optimize the detection of minor clones and therefore improve the prognostic classification of patients, in accordance with the mentioned studies.

Regarding *FLT3*-ITD mutation as a biomarker of MRD in patients undergoing allo-HSCT, many authors rule it out as an unstable mutation during the course of the disease [22,23]. However, several studies that analyzed *FLT3*-ITD using different methodologies, such as RT-qPCR or NGS, suggest that it could be a useful marker, despite its instability [24,25,26,27]. In the present study, the *FLT3*-ITD mutation analysis in DNA and cDNA samples was performed at specific time points pre- and post-allo-HSCT. With regard to performing the analysis before relapse, *FLT3*-ITD expression seemed to be very useful in predicting relapse. In some patients, the *FLT3*-ITD cDNA measurement did not anticipate the relapse due to the loss of the mutation or because the prior sample was obtained too early to detect the mutation. Despite these issues, which can be solved by complementing this technique with other MRD markers, such as *WT1* or *NPM1* RT-qPCR, chimerism or MFC, the assessment of *FLT3*-ITD expression by capillary electrophoresis could be an excellent approach to MRD measurement after allo-HSCT. In addition, the patients in whom the mutation is detected could benefit from FLT3i treatment in order to eradicate the resistant clones harboring the *FLT3*-ITD mutation.

FLT3is have been shown to improve outcomes in patients with *FLT3*-mutations, and they are included in AML clinical management algorithms [28,29]. In this sense, MRD monitoring is important for determining the efficacy of the inhibitor. Some studies have employed a PCR-NGS assay in DNA samples for evaluating the clearance of *FLT3*-ITD mutations in patients who underwent an FLT3i [27,30]. Despite its high sensitivity, this approach could be laborious and expensive, so it may not be available for everyone. Therefore, after assessing the usefulness of *FLT3*-ITD mutation expression as a biomarker during allo-HSCT follow-up, we also evaluated its usefulness after FLT3i treatment. All patients but one were refractory to FLT3i. This one patient achieved complete remission, confirmed by the absence of *FLT3*-ITD mutations in the cDNA samples. These results suggest that *FLT3*-ITD expression measurement could be a good biomarker for treatment effectiveness.

## 5. Conclusions

Our results suggest that the cDNA fragment analysis of *FLT3*-ITD mutation by capillary electrophoresis is an easy-to-implement technique that could be a useful alternative approach in patients with AML at diagnosis, during allo-HSCT monitoring and in post-FLT3i follow-up. However, these findings need to be confirmed in studies with larger numbers of patients.

## Figures and Tables

**Figure 1 cancers-14-04006-f001:**
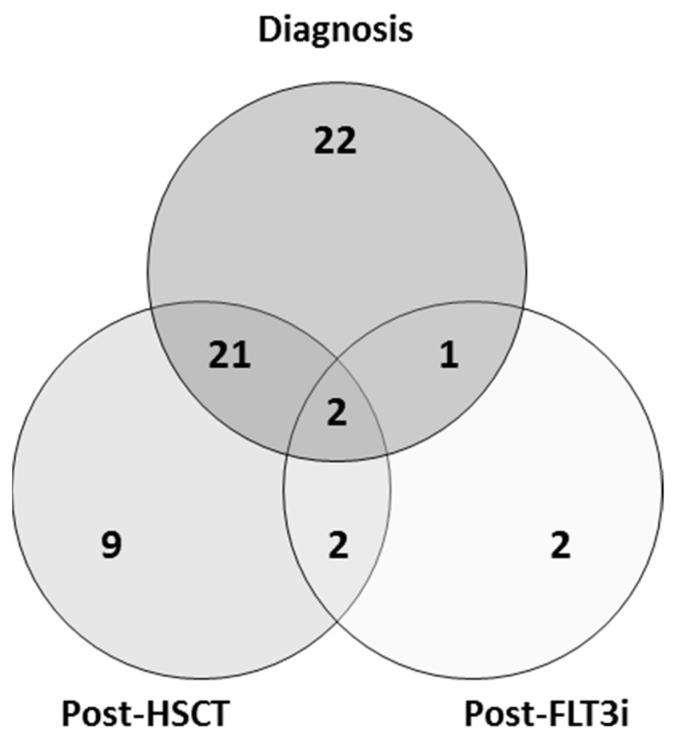
Venn diagram showing the distribution of patients within the cohorts of the study. Allo-HSCT: allogeneic hematopoietic stem cell transplantation. FLT3i: tyrosine kinase inhibitor.

**Figure 2 cancers-14-04006-f002:**
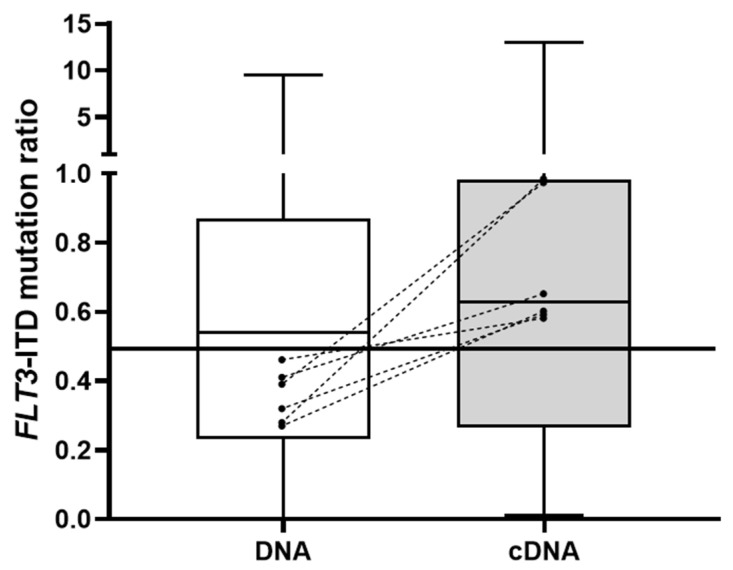
Boxplots showing the comparison of *FLT3*-ITD mutation AR between DNA and cDNA samples. The six cases in which ELN risk stratification changed from intermediate-risk to high-risk AML in cDNA are represented by dashed lines.

**Figure 3 cancers-14-04006-f003:**
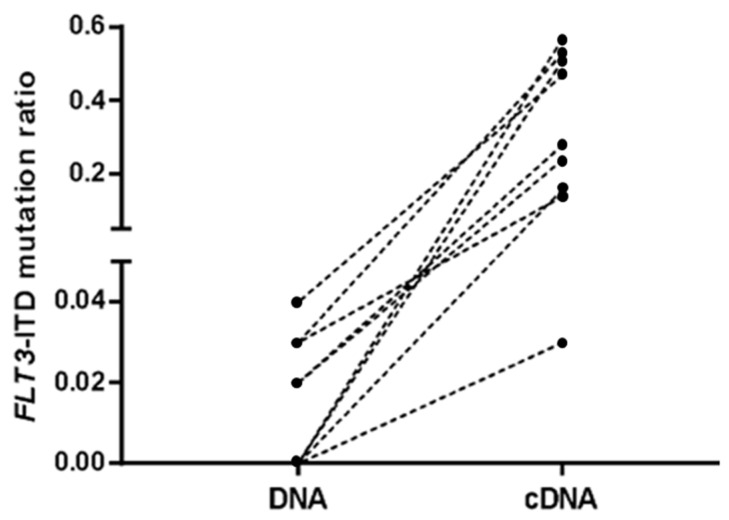
Comparison of *FLT3*-ITD mutation ratio between DNA and cDNA in the samples immediately prior to progression of the 12 patients who relapsed. *FLT3*-ITD mutations were undetectable in three patients in their DNA and cDNA samples.

**Figure 4 cancers-14-04006-f004:**
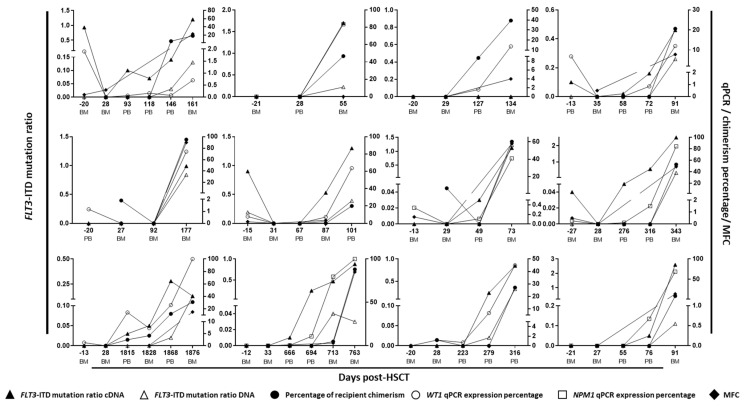
Follow-up monitoring of the 12 patients who relapsed from post-allo-HSCT cohort, including DNA and cDNA *FLT3*-ITD mutation, chimerism and *NPM1* and *WT1* expression analysis. MFC: multiparametric flow cytometry. HSCT: hematopoietic stem cell transplantation. BM: bone marrow sample. PB: peripheral blood sample.

**Figure 5 cancers-14-04006-f005:**
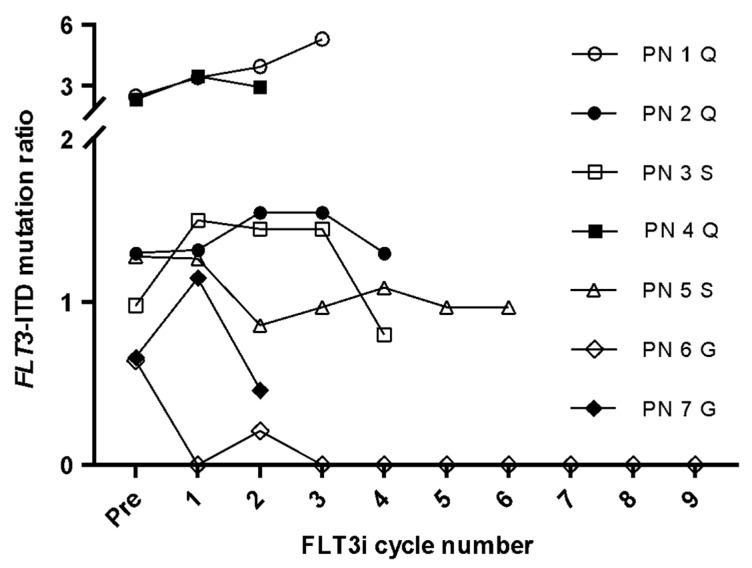
*FLT3*-ITD mutation ratio during follow-up of the seven patients treated with *FLT3* tyrosine kinase inhibitors. PN: patient number. FLT3i: tyrosine kinase inhibitors. Q: quizartinib. S: sorafenib. G: gilteritinib.

**Table 1 cancers-14-04006-t001:** Demographic and clinical characteristics of patients of the three cohorts. Allo-HSCT: allogeneic hematopoietic stem cell transplantation. FLT3i: tyrosine kinase inhibitor. IA: idarubicine and cytarabine. R/R AML: refractory/relapsed acute myeloid leukemia.

Cohort	Diagnosis	Post-HSCT	Post-FLT3i
n	46	34	7
Age, median (range), years	63 (27–91)	45 (27–65)	52 (31–65)
Female sex, n (%)	15 (32.6)	14 (41.2)	3 (42.9)
Induction therapy, n (%)			
IA 3 × 7	27 (58.7)	29 (85.3)	5 (71.4)
IA 3 × 7 + FLT3i	5 (10.9)	2 (5.9)	2 (28.6)
Hypomethylating	3 (6.5)	0 (0.0)	0 (0.0)
Other	1 (2.1)	3 (8.8)	0 (0.0)
Palliative care	10 (21.8)	0 (0.0)	0 (0.0)
Allo-HSCT			
HSCT type, n (%)			
Haploidentical	-	17 (50.0)	-
HLA-identical	-	15 (44.1)	-
Haplo-cord	-	2 (5.9)	-
Conditioning regimen, n (%)			
Myeloablative	-	26 (76.5)	-
Reduced intensity	-	8 (23.5)	-
FLT3i in patients with R/R AML			
Quizartinib	-	-	3 (42.8)
Sorafenib	-	-	2 (28.6)
Gilteritinib	-	-	2 (28.6)

## Data Availability

The data presented in this study are available on request from the corresponding author.

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
