# Peer review of "FLT3*-ITD Expression as a Potential Biomarker for the Assessment of Treatment Response in Patients with Acute Myeloid Leukemia"

_cancers, 2022, doi:10.3390/cancers14164006_

Round 1
Reviewer 1 Report
Carbonell at al. analyzed the value of using FLT3-ITD expression levels in addition or instead of quantifying the allelic ratios of FLT3-ITD and the wild-type allele on genomic DNA. They show that FLT3-ITD mutation analysis using cDNA has a higher sensitivity and suggest that it may be used as additional biomarker for the assessment of treatment response and prediction of an impeding relapse.
Overall the study is well-conducted but its novelty is limited. I have some suggestions to increase its impact and to gain further insights into the unresolved question why the measurement of the allelic ratios - at least in some cases - substantially differs when assessed on DNA or cDNA.
The comparison of FLT3-ITD analysis using genomic and cDNA at diagnosis is not new and the limited number of patients analyzed adds little to the current knowledge. The data regarding the monitoring of the FLT3-ITD mutation burden in the post-allo transplantation and FLT3i treatment setting might be novel; however, given the obvious higher sensitivity of measuring FLT3-ITD allelic ratios on cDNA, they are not really surprising.
As mostly only patient numbers are provided, the data are rather difficult to follow, the percentages of positive or negative patients (in addition to the actual numbers) throughout the manuscript would be helpful to better grasp potential significant differences between the two methods.
A true added value would be the analysis of the (amplification) efficiency of the employed assays - DNA vs cDNA - to identify the reasons for the differing allelic ratios and detectable mutation burden. A further analysis of the samples lacking expression of the wild-type FLT3 allele is of major interest. The question whether this is the case because the second allele is deleted or due to the fact that the non-mutated allele is not expressed is important to fully understand the differences in mutation detection between DNA and cDNA.
The discussion is rather lengthy and might benefit from shortening and reducing it to the most important findings omitting speculations without sufficient evidence.
Author Response
Dear Reviewer
First of all, thank you very much for your time and for reviewing our manuscript.
We have modified the manuscript based on the suggested changes.
In addition, The manuscript has been revised by a professional translator. (Morote traducciones, UNE EN ISO 9001:2015 Certified, UNE EN ISO 17100:2015 Certified).
The following is a summary of the responses to the suggestions:
- As mostly only patient numbers are provided, the data are rather difficult to follow, the percentages of positive or negative patients (in addition to the actual numbers) throughout the manuscript would be helpful to better grasp potential significant differences between the two methods.
Percentages were added to the text along with the number of FLT3-ITD positive and negative patients.
- A true added value would be the analysis of the (amplification) efficiency of the employed assays - DNA vs cDNA - to identify the reasons for the differing allelic ratios and detectable mutation burden.
The amplification efficiency of both assays did not differ. For this reason, differences observed were probably due to the presence of more quantity of RNA copies in the sample, comparing to DNA copies.
- A further analysis of the samples lacking expression of the wild-type FLT3 allele is of major interest. The question whether this is the case because the second allele is deleted or due to the fact that the non-mutated allele is not expressed is important to fully understand the differences in mutation detection between DNA and cDNA.
We added in the text: "Although the wild-type alleles were detected in DNA, there appeared to be no normal allele expression in these two cases, which would have a similar effect to a loss of heterozygosity. Loss of the wild-type allele has been associated with poorer overall survival and progression-free disease. [21] In the case of these two patients, both had an unfavorable clinical course, confirming this prognosis". (Page 8, 248-253). We also added the reference 21, which supports the bad prognosis of patients without FLT3 wilt type allele.
- The discussion is rather lengthy and might benefit from shortening and reducing it to the most important findings omitting speculations without sufficient evidence.
Discussion was shortened, omitting pre-allo-HSCT and day 30 post-allo-HSCT data, which not added to much information, due to the low number of events.
Reviewer 2 Report
1. Please provide more detail regarding the 46 patients, including WBC count at diagnosis and the presence of other genetic lesions, including NPM1, CEBPA, UBTF, etc.
2. If possible, provide the results of flow cytometric analyses performed on the bone marrow aspirates at the times that FLT3 was analyzed.
Author Response
Dear Reviewer
First of all, thank you very much for your time and for reviewing our manuscript.
We have modified the manuscript based on the suggested changes.
In addition, The manuscript has been revised by a professional translator. (Morote traducciones, UNE EN ISO 9001:2015 Certified, UNE EN ISO 17100:2015 Certified).
The following is a summary of the responses to the suggestions:
- Please provide more detail regarding the 46 patients, including WBC count at diagnosis and the presence of other genetic lesions, including NPM1, CEBPA, UBTF, etc.
We have created the Supplementary Table 1 (Table S1), which contains information about gender, age, white blood cell count, platelet count, hemoglobin, FLT3-ITD allele ratio in DNA and cDNA samples, mutations in NPM1, IDH1, IDH2, CEBPA and DNMT3A; and karyotype of the 46 patients at diagnosis.
- If possible, provide the results of flow cytometric analyses performed on the bone marrow aspirates at the times that FLT3 was analyzed.
Multiparametric flow cytometry data of bone marrow samples during follow-up was added in the Figure 4 and cited in the text in Methods section.
Reviewer 3 Report
Regarding to the manuscript: “FLT3-ITD expression as a potential biomarker for the assessment of treatment response in patients with acute myeloid leukemia”:
The authors aim to compare FLT3-ITD mutation analysis in DNA and cDNA samples at diagnosis and to prove the usefulness of its expression measurement as MRD marker after allogeneic stem cell transplantation (allo-HSCT) or FLT3 inhibitor (FLT3i) administration.
I consider this manuscript is a relevant work in the field, since it provides results that contribute to the discussion about the use of cDNA as a sample for the detection of FLT3-ITD mutations and its use in both diagnosis and prognosis of patients with AML. However, there are several points in the manuscript that need to be improved/reviewed for publication mainly in the discussion section.
-The wording of the manuscript should be improved. An English review by an expert must be carried out. In addition, there is omitted information that may be relevant to researchers who are not experts in the field. For example, highlighting the cut-off values for AR at diagnosis and MDR would be valuable for the work.
-In the introduction (line 57), author affirm that “(...) the only curative treatment for high-risk AML patients is allogeneic hematopoietic stem cell transplantation (allo-HSCT), (…)”. This is a strong statement, so it should be accompanied by a quote that primarily focus on allo-HSCT treatment in AML or indicating that it is the curative option for most of R/R patients.
-In the introduction (lines 73-74), the aims of the work are not clear enough. Authors could avoid repeating the word "implication" and be more specific in what they intend to assess.
-In the results section 3.3 (lines 201-207): Why was the comparison between cDNA and DNA not performed for this group of patients? It should be indicated if the mutation was detected from cDNA or DNA
-In the Discussion (lines 224-226), the reference 19, which was performed exclusively on adult patients, seems to be masked, due to the way the sentence is written. I suggest rephrasing the sentence. Because in the mentioned work (19), the comparison between cDNA and DNA is also carried out in a large number of samples from adult patients with AML, I suggest to include it in the discussion of the results.
- In the Discussion (lines 234-235), if there is a good correlation between the results obtained when the analysis was carried out on DNA and cDNA samples, it is not clear to me why it is concluded that the analyzes are not equivalent (line 239). I suggest rephrasing this paragraph.
- In the Discussion (line 255), the bibliographic citations to which reference is made must be indicated
- In order to conclude that cDNA detections could be an alternative, a larger number of samples should be analyzed. This should be highlighted in concussion.
Author Response
First of all, thank you very much for your time and for reviewing our manuscript.
We have modified the manuscript based on the suggested changes.
The following is a summary of the responses to the suggestions:
-The wording of the manuscript should be improved. An English review by an expert must be carried out.
The manuscript has been revised by a professional translator. (Morote traducciones, UNE EN ISO 9001:2015 Certified, UNE EN ISO 17100:2015 Certified).
- There is omitted information that may be relevant to researchers who are not experts in the field. For example, highlighting the cut-off values for AR at diagnosis and MDR would be valuable for the work.
We added in the text: “AMLs with allelic ratios >0.5 were classified as high-risk AML, according to the ELN 2017 risk stratification algorithm. Cut-off values for FLT3-ITD AR by capillary electrophoresis were 0.03 for diagnosis samples and 0.01 for MRD samples”. Page 4, lines 123-127
-In the introduction (line 57), author affirm that “(...) the only curative treatment for high-risk AML patients is allogeneic hematopoietic stem cell transplantation (allo-HSCT), (…)”. This is a strong statement, so it should be accompanied by a quote that primarily focus on allo-HSCT treatment in AML or indicating that it is the curative option for most of R/R patients.
The statement was rephrased and a new reference on allo-HSCT was added (reference 5). Page 2, lines 57-59.
-In the introduction (lines 73-74), the aims of the work are not clear enough. Authors could avoid repeating the word "implication" and be more specific in what they intend to assess.
We have reformulated the objectives. Page 2, lines 73-77.
-In the results section 3.3 (lines 201-207): Why was the comparison between cDNA and DNA not performed for this group of patients? It should be indicated if the mutation was detected from cDNA or DNA
We indicated in the text the reason why we only performed the cDNA analysis in FLT3i cohort. Page 7, lines 211-213.
-In the Discussion (lines 224-226), the reference 19, which was performed exclusively on adult patients, seems to be masked, due to the way the sentence is written. I suggest rephrasing the sentence. Because in the mentioned work (19), the comparison between cDNA and DNA is also carried out in a large number of samples from adult patients with AML, I suggest to include it in the discussion of the results.
We clarified which studies about FLT3-ITD were performed in pediatric and adult cohorts: “Analysis of cDNA samples by capillary electrophoresis to detect FLT3-ITD mutations at diagnosis has been described in a few studies, mainly in pediatric patients [18, 19] and, to the best of our knowledge, in only one study in adult patients [20]". Lines 235-237.
- In the Discussion (lines 234-235), if there is a good correlation between the results obtained when the analysis was carried out on DNA and cDNA samples, it is not clear to me why it is concluded that the analyzes are not equivalent (line 239). I suggest rephrasing this paragraph.
Although there is a good correlation between the two analyses, it is not perfect (since it is not equal to 1), as the values differ significantly. Therefore they cannot be equivalent. However, to avoid possible misunderstandings, we rephrased the paragraph. Page 8, lines 244-247.
- In the Discussion (line 255), the bibliographic citations to which reference is made must be indicated
Due to the suggestion of another Reviewer #1, we shortened the discussion to the most important findings omitting pre-allo-HSCT and day 30 post-allo-HSCT findings, since there is not enough evidence of its usefulness. For this reason, that sentence was eliminated.
- In order to conclude that cDNA detections could be an alternative, a larger number of samples should be analyzed. This should be highlighted in concussion.
We added the sentence: “However, these findings need to be confirmed in studies with larger numbers of patients” in the conclusions section. Page 9, lines 289-290.
Round 2
Reviewer 1 Report
The authors have properly addressed my concerns and the revision of the manuscript has significantly improved its clarity. There are a few minor typos, which need to be corrected, e.g. line 99: tipically / typically.
Reviewer 3 Report
The authors have answered all my questions and suggestions satisfactorily.